# Individual Locating of Soccer Players from a Single Moving View

**DOI:** 10.3390/s23187938

**Published:** 2023-09-16

**Authors:** Adrien Maglo, Astrid Orcesi, Julien Denize, Quoc Cuong Pham

**Affiliations:** Université Paris-Saclay, CEA, List, F-91120 Palaiseau, France; julien.denize@cea.fr (J.D.); quoc-cuong.pham@cea.fr (Q.C.P.)

**Keywords:** sports field registration, team sport players tracking, neural networks, computer vision

## Abstract

Positional data in team sports is key in evaluating the players’ individual and collective performances. When the sole source of data is a broadcast-like video of the game, an efficient video tracking method is required to generate this data. This article describes a framework that extracts individual soccer player positions on the field. It is based on two main components. As in broadcast-like videos of team sport games, the camera view moves to follow the action and a sport field registration method estimates the homography between the pitch and the frame space. Our method estimates the positions of key points sampled on the pitch thanks to an encoder–decoder architecture. The attention mechanisms of the encoder, based on a vision transformer, captures characteristic pitch features globally in the frames. A multiple person tracker generates tracklets in the frame space by associating, with bipartite matching, the player detections between the current and the previous frames thanks to Intersection-Over-Union and distance criteria. Tracklets are then iteratively merged with appearance criteria thanks to a re-identification model. This model is fine-tuned in a self-supervised way on the player thumbnails of the video sample to specifically recognize the fine identification details of each player. The player positions in the frames projected by the homographies allow the obtaining of the real position of the players on the pitch at every moment of the video. We experimentally evaluate our sport field registration method and our 2D player tracker on public datasets. We demonstrate that they both outperform previous works for most metrics. Our 2D player tracker was also awarded first place at the SoccerNet tracking challenge in 2022 and 2023.

## 1. Introduction

With the growing popularity of sports analytics in recent years, there has been recent interest in player positional data. Knowing the position of each player on the field is, indeed, a key to evaluate the players’ individual and collective performances. A solution for outdoor sports is to equip the players with GPS devices. Radio solutions are also available for indoor sports but they require the arenas to be setup with specific receivers. Moreover, the positional data are, most of the time, kept private by the teams. This prevents analysis by the opposing team. Thus, video tracking becomes the sole solution for obtaining these data.

Tracking team sport players in videos has been an active research topic. It is a multiple object tracking (MOT) problem with specific challenges. Players on the same team, indeed, share a very similar appearance since they wear the same jerseys. They move quickly and can adopt various poses, which can make their detection difficult. Some play events, such as soccer corners, generate a high number of occlusions. The tactical camera mainly used in broadcast video has a large field of view that allows the visualization of a large part of the pitch. However, the players are small in the images (about 100 pixels high and 50 pixels wide in standard HD videos). The access to fine identification details is consequently limited. The camera view can also move quickly to follow the action. Player detection and tracking are then made complex by the motion blur and the large moves of the players’ bounding boxes. Figure 1 illustrates some of these challenges for soccer.

In this paper, we present a new framework that can extract the individual players’ positions from a broadcast-like video of a soccer game. Although commercial products [2] have been proposed to achieve this task, as far as we know, such systems, their limitations, and the quantitative evaluation of their performance have not been described in the literature. Our framework is composed of three main components, as depicted in Figure 2:Since, in a broadcast-like video, the tactical view moves, the first component is a soccer field registration method. This determines the position of the soccer pitch in each video frame. The homography between the pitch space and the frame space is computed by estimating the positions of key points sampled over the pitch. One of the primary challenges of the registration stage is to process different partial views of the field, with differing visual characteristics, in order to correctly estimate the homography regardless of the specific view.The second component is a 2D player tracking algorithm. It generates tracklets in the frame space. The major challenge for robust player tracking is to maintain the correct identity of each player over time, despite variability in pose and resolution, frequent occlusions, and temporary exits from the field of view.By projecting the results of the 2D tracking on the pitch space thanks to the estimated homographies, our framework outputs the individual positions of the players on the pitch.

Our contributions cover each of the following components:We propose a new sports field registration method that better captures relevant visual features distributed across the entire image. Our encoder–decoder model is based on a vision transformer and generates heatmaps to locate the pitch keypoints. Previous sports field registration methods use models based on convolutions [3,4,5,6,7,8,9,10,11,12,13], which are limited by their receptive fields. The attention mechanisms of our vision transformer encoder [14] can capture characteristic pitch features globally in the frames. Instead of using a uniform distribution of the pitch keypoints in the pitch space [10,11], we applied a perspective-aware sampling of the keypoints. This produces a more uniform distribution in the frame space to improve the homography estimation using the RANSAC algorithm. The pre-training of sports field registration models was not studied in the literature. We demonstrate that our self-supervised pre-training of the encoder improves the adaptation of our model to the soccer images’ target domain.We present a robust player tracking approach that implements a low-noise tracklet generation and fusion strategy, and an online learning method for fine-grained re-identification. Our method starts by associating, with bipartite matching, the player detections of successive frames with Intersection-Over-Union (IoU) and distance criteria. The number of player identity switches is reduced by compensating the camera motion between the frames. This quick method generates good quality tracklets between the instances when players occlude each other. For the re-identification (re-ID) of the players, previous methods use a frozen convolutional neural network during the tracking [15,16,17,18]. These approaches have difficulties in recognizing the fine identification details of the players visible in the video clips. Our approach specializes the re-ID network on each clip by fine-tuning it with a triplet loss on the previously generated tracklets. Tracklets are then associated on an appearance criteria with an iterative merging algorithm in order to generate complete tracks. It starts with the tracklets that look the most similar and stops when the matching candidates are too dissimilar.We demonstrate the performance of the proposed methods through extensive experiments on several datasets. Our homography estimation method is state-of-the-art and outperforms previous works on multiple metrics using the WorldCup [3] and the TS-WorldCup [11] public datasets. Our tracking algorithm also outperforms previous works on the SoccerNet-tracking [1] and SportsMOT [19] public datasets.

The first section of this article is dedicated to the state-of-the-art in terms of sports field registration and team sport player tracking. In the second section, we describe our methods. The third section is dedicated to the experimental results. We separately demonstrate the efficiency of our sports field registration method and our tracking method. We also combine them to generate individual player trajectories. In the fourth section, we discuss our findings before concluding in the sixth section.

## 2. Previous Work

### 2.1. Sports Field Registration

In order to accurately determine the positions of players in the field space, it is essential to localize the soccer pitch within the input frame when the camera field of view adjusts to track the action. Previous studies in the literature have addressed the registration of sports fields in videos, with some focusing on utilizing initial annotations [20,21,22] and others without such annotations [23]. This subsection specifically emphasizes the exploration of a fully automatic approach for sports field registration.

#### 2.1.1. Approaches Based on Handcrafted Features

The initial approaches for registering sport fields use conventional image operators to detect the lines and ellipses on the field, such as Hough transforms [24] or luminance thresholding with a RANSAC algorithm on pixels [25]. The detected lines are then associated with the model lines with a combinatorial optimization [24,25] or a probabilistic decision tree [26]. The position of line intersections then allows the computing of the final homography. Yao et al. [27] do not classify the soccer pitch lines but match the intersections between them to 16 patterns. An exhaustive homography evaluation is performed by back-projecting the pitch model to the frame space. Methods based on hand-crafted line detectors, however, suffer from difficult tuning in working with various capture conditions (colors, contrast, luminosity, etc.).

#### 2.1.2. Deep Learning Approaches

Some recent sports field registration approaches leverage the deep neural network’s ability to generate high-quality segmentation results. The process of these methods can be summed up into three steps. The first step uses a deep neural network to segment the field elements (lines or zone). The second step matches the segmented elements of the input frame with the pitch model elements to generate the homography. Finally, the results are refined in a third step to improve the homography quality.

For the segmentation networks, various models have been used: a VGG-16 model [28] by Homayounfar et al. [3], a U-Net by Sha et al. [5], and a DeepLabV3 ResNet [29] by Theiner and Ewert [13]. Other methods resorted to Generative Adversarial Networks to learn a translation between real and pitch model images. Thus, Sharna et al. [4], Chen and Little [8], and Zhang and Izquierdo [9] use the pix2pix framework [30].

Various techniques can match the segmented elements with the pitch model element. Homayounfar et al. [3] ran a branch and bound algorithm on a Markov random field. Sharna et al. [4], Sha et al [5], and Chen and Little [8] generate features and match them with the closest model feature thanks to a $L^{2}$ distance. Ciopa et al. [31] distilled the knowledge of a teacher model to a student network. TVCalib [13] estimates the full camera calibration by minimizing a differentiable objective function that represents the segment reprojection error. Zhang and Izquierdo [9] estimate the position of four control points with a second network. The final homography is obtained thanks to a DLT algorithm [32].

Regarding the refinement strategies, Sharna et al. [4] resort to a Markov random field optimization. Sha et al. use a spatial transformer network [5], and Chen and Little a Lucas–Kanade algorithm [33].

Other sports field registration methods estimate the homography by directly locating key points without preliminary segmentation. Thus, the frameworks of Jiang et al. [6] and Fani et al. [34] regress the real-world positions of four control points in the frame space. In the method of Jiang et al. [6], the initial estimation is refined by minimizing the re-projection error. Instead of estimating the positions of frame key points in the basketball court space, Citaro et al. [7] estimate the positions of court corners in the frame space with a U-Net network. Nie et al. [10] do not use key points located on soccer pitch corners but sample them regularly on a grid. Even if these key points are not located on pitch interest points, the receptive field of the convolutional neural network can capture their features, sometimes far from them. A RANSAC algorithm is used for the homography estimation in order to be robust to misestimated key point locations. The regular sampling of the key points in the pitch space may, however, favor the selection by the RANSAC of key points mostly far from the camera. These key points are, indeed, closer to each other in the frame space due to the effect of perspective. This consequently leads to a sub-optimal homography estimation. This method has been recently improved by Chu et al. [11] by the integration of dynamic filters in the encoder–decoder network. Jacquelin et al. [12] also build upon this framework for swimming pool registration.

#### 2.1.3. Public Datasets

The abundance of data is key to training a sports field registration method based on a deep neural network. Many of the previously introduced works resorted to private datasets for training and testing their method. Some authors, however, released their dataset publicly, which makes comparisons easier. For soccer, the WorldCup dataset [3] is often used as a reference in the literature for soccer pitch homography estimation from a tactical perspective. The bigger TS-WorldCup dataset [11] is composed of 43 video clips. The SoccerNet project offers a substantial dataset for camera calibration [35]. However, this dataset includes images from various viewpoints, including fisheye views from inside the goal. These views are highly distorted, rendering it unfeasible to generate a homography between the field and frame space. Jaquelin et al. [12] released a swimming pool registration dataset with their method. The DeepSportRadar dataset [36] has also been released for basketball camera calibration. Table 1 summarizes the public sport field registration datasets.

### 2.2. Player Tracking

An efficient player tracking system is required to detect and track the players in all video frames so as to later determine their position on the soccer pitch. Team sport players tracking is a specific sub-task of multiple people tracking or multiple object tracking (MOT). The development of multiple people tracking frameworks has been mainly driven by video surveillance applications. Public challenges were proposed to evaluate the approaches to common datasets. One significant example is the MOT16 challenge [37]. Its training set incorporates busy indoor and outdoor urban scenes filmed with a single static or a moving camera carried by a vehicle or a pedestrian. Nevertheless, team sport players tracking has its own characteristics that motivate the design of specific approaches. Team-sport players move quicker than pedestrians walking in a street or a mall. Within a team, players share a very similar appearance since they wear the same jerseys. They also often exit the camera field of view to later re-enter. In the first part of this subsection, we briefly review multiple people tracking frameworks before, in the second part, thoroughly reviewing multiple player tracking methods.

#### 2.2.1. Multiple People Tracking

MOT algorithms can be classified into two main categories: online methods and offline methods.

Offline methods take advantage of the complete sequence of images. These methods often adopt a graph-based approach [38,39], where detections in each frame correspond to vertices. The edges link the detections together to form tracks. They are globally optimized using a minimum cost flow iterative algorithm [38] or a k-shortest path formulation solved with linear programming [39]. Hornakova et al. [40] added lifted edges to the graph in order to model the long-term connections between detections. The approach of Brasó and Leal-Taix [41] trains a fully differentiable network that extracts features from each detection and estimates the probability of an existing connection between them.

To tackle real-time scenarios, online methods associate new detections with existing tracks using only the current and past frames. To match new detections with existing tracks, the SORT algorithm [42] uses a Kalman filter [43] to predict the future positions of existing tracks. Then, the Hungarian algorithm matches the current frame detections with the prediction thanks to an IoU criteria. ByteTrack [44] introduces a two-step association algorithm, focusing on high-confidence detections in the first step and handling low-confidence ones in the second step. More recently, OC-SORT [45] interpolates the missing detections of tracks recovered after being lost to better estimate the parameters of the Kalman filter. It also adds a motion direction component to the input of the Hungarian algorithm. The Deep SORT algorithm [46] incorporates a visual appearance term in the association criteria that consists in a cosine distance between re-ID vectors. BoT-SORT [47] extends Bytetrack [44] by also adding re-ID criteria. High-confidence detections are matched with a combination of IoU and appearance criteria. TransMOT [48] computes the appearance features and matches the detections between frames with Transformer architectures.

Employing separate networks for detection and re-ID offers the advantage of addressing the opposite objectives of each task. The detection network focuses on learning shared features to recognize humans, while the re-ID network aims to learn unique features for each individual. However, this approach can introduce scalability challenges, as each detected bounding box requires independent processing by the re-ID network. To mitigate these scalability concerns, single-shot methods have been proposed. These methods utilize a single network to generate both the bounding box coordinates and the corresponding re-ID vectors [49]. Thus, JDE [50] uses a FPN architecture [51] with multiple prediction heads at three levels to learn the detection and re-ID tasks. The reciprocal network of CSTrack [52] has been integrated into JDE to better manage the opposite objectives of the detection and re-ID tasks. Zhang et al. propose an anchor-free approach named FairMOT [53]. Track-RCNN [54] adds tracking to Mask-RCNN [55], a detection and segmentation method. Pang et al. [56] densely match the detector region proposals of the current and previous frame for the contrastive learning of the appearance features. Trackformer [57] uses a Transformer architecture to detect and match the detections.

Traditional multiple object tracking (MOT) methods often result in frequent identity switches when applied to team sport players. Whenever players go out of the field of vision or remain occluded for a significant duration, new identities are assigned to them when they reappear. The varied poses of the players also hinder their detection. Finally, the fast motion of the players and the camera is also a challenge for the prediction of bounding box positions. Consequently, traditional MOT algorithms have difficulties in achieving a good performance on team sports datasets without modifications (see Section 4.2).

#### 2.2.2. Team Sport Players Re-Identification

The difficulty of team sport player tracking is that, most of the time, the players from the same teams wear the same jersey. One way to distinguish them is to use jersey numbers. The first jersey number recognition methods, such as the framework of Ye et al. [58], have been developed around handcrafted features such as Zernike moment features [59]. The most recent works are, however, based on convolutional neural networks [60,61]. To improve the number classification accuracy, some methods resorted to the preprocessing of the player thumbnails. Thus, the approach of Li et al. [62] focuses on the correction of the number distortion. The method of Liu and Bhanu [63] estimates the players pose in order to precisely locate their jersey numbers. Another strategy is to leverage the temporal dimension. The approach of Chan et al. [64] performs jersey number predictions on player tracklets with a ResNet + LSTM network [65,66].

Jersey numbers are, nevertheless, not visible from any camera view point. In many cases, appearance details are the only way to re-identify players. Re-identification models have been successfully utilized in the context of basketball. Indeed, the size of the play field makes the players’ thumbnail resolution often important enough to access fine identity traits. The framework of Teket and Yetik [67] is based on a MobileNetV2 [68] network, trained with a triplet loss. Senocak et al. [69] combine part-based features and multiscale global features. CLIP-ReIdent [70] leverages the pretraining of a Contrastive Language–Image Pretraining (CLIP) model [71] with a contrastive image-to-image training approach.

#### 2.2.3. Team Sport Players Tracking

Some MOT frameworks have been specifically designed to track team sport players. Manafifard et al. propose a review article about soccer player tracking [72]. Background subtraction algorithms, Haar features [73] or a deformable part model [74] have been used the soccer player detection [75,76,77] or basketball player detection [78]. The matching of the bounding boxes of each frame relies, for most approaches, on position, motion, and appearance features based on color histograms. It is formulated as a Markov chain Monte Carlo data association [76], a merge split strategy [75] or particle filtering [77].

Recent approaches leverage deep neural networks to detect the players. For example, Theagarajan and Bhanusome [15] use YOLOv2 [79] to detect soccer players, while Zhang et al. [80] use Mask R-CNN [55]. Hurault et al. [16], Vats et al. [81], and Maglo et al. [82] use a Faster R-CNN model [83] to detect soccer, hockey, and rugby players. The YOLOX detector [84] has also been used in many recent frameworks [17,18,85] since it offers a state-of-the-art trade-off between detection speed and accuracy.

Generic MOT algorithms have been utilized as the base of frameworks targeting sport data. For example, the DeepSORT tracker [46] is used in the approach of Theagarajan and Bhanu [15]. The K-shortest paths formulation [39] is used by Zhang et al. [80] for their multi-camera soccer player tracker. Yang et al. [86] bring iterative refinements to the approach of Zhang et al. with a loss at the image level. The matching of new detections to existing tracks in the approach of Hurault et al. [16] is based on a spatial appearance criteria learned in a self-supervised way with a triplet loss. The approach of Kong et al. [87] is based on the players’ poses modeled by a LSTM network. For Hockey, Vats et al. [81] resorted to an offline tracker [41] and two ResNet-18 networks [65] to identify teams and jersey numbers. In our previous work [82], we used a Transformer network to iteratively match the tracklets of rugby players with a single appearance criteria thanks to few user annotations. Yang et al. [85] proposed to expand the detected bounding boxes of soccer players to better match them between frames. Huang et al. [17] combined the OC-SORT algorithm [45] with the OSNet re-identification network [88] and sport-specific tracklet-matching strategies. Wang et al. [18] use three Hungarian matching algorithms that mix IoU and re-identification criteria. They restore the tracks of players that exit and re-enter with a re-identification criteria.

A great challenge still remains in the disambiguation of the players from the same team, especially when their bounding boxes are very small in the video. The design of efficient re-identification methods that can leverage fine details, visible on the scale of a few pixels, is essential to achieve an overall good player tracking performance.

#### 2.2.4. Public Datasets

The availability of public team sport player tracking datasets is key to allowing model training and performance comparisons between approaches. The APIDIS dataset [89] provides tracking data for one minute of a basketball game with multiple static views. For the tracking of soccer players, the FIGC-CNR dataset [90] provides tracking data for multiple static views. The larger SoccerNet tracking dataset [1] offers tracking data for a single moving tactical view. SportsMOT [19] provides tracking data for 240 video clips of soccer, basketball, and volleyball. Maglo et al. [82] also released a public dataset for rugby sevens player tracking for a single moving tactical view. Table 2 summarizes the content of these datasets.

## 3. Method

This section describes the three main components of our approach as depicted in Figure 2. Our sports field registration method generates homographies between the pitch and the frame space for each frame. Our 2D tracker provides the bounding boxes of each player in the frame space along with a constant tracking identifier. The projection module takes these two outputs to finally compute the players’ positions in the pitch space.

### 3.1. Sports Field Registration

#### 3.1.1. Overview

Our sports field registration method is built upon the framework of Nie et al. [10]. Compared to other methods, the latter has the advantage of having fast inference times and being simple to implement. However, it can fail to generate accurate homographies in cases where only a few pitch features are visible in the image or when the camera moves quickly. We improve the accuracy over the original method by Nie et al. and others inspired by it, by making several significant modifications: a new network architecture, a perspective-aware key point sampling strategy, self-supervised encoder pre-training, and data augmentation techniques. To estimate the homography between the pitch space and the frame space, an encoder–decoder model computes score heat maps for each of the key points positioned on the pitch template, as depicted in Figure 3. The heat maps’ centers of mass provide the positions of the key points. The homography is then estimated using a RANSAC algorithm [91] with DLT [32] in order to be robust to wrong key point positions.

#### 3.1.2. Network Design

Our model generates a heat map for each of the K different key points sampled from the court template and a heat map for the background areas.

Nie et al. use a ResNet-18 [65] encoder with dilated convolution, non-local blocks as the encoder, and four deconvolution layers as the decoder. Chu et al. [11] use a ResNet-34 as the encoder and four up-sampling blocks as the decoder. We propose a new encoder–decoder architecture. Our encoder is a vision transformer (ViT-tiny) [14]. Vision transformers have demonstrated their ability to achieve superior performance in various computer vision tasks. Indeed, vision transformers, with their attention mechanism, allow a global analysis of the image in a more efficient way than using multiple convolutions to progressively increase the receptive field. The co-occurrence and the relative positioning of the pitch features are key to accurately locate the keypoints. Furthermore, we want the key points’ locations to be based on local but also global features in order to be more robust to occlusions. We choose the tiny version of the ViT since it achieves a good trade-off between accuracy and inference speed. Our lightweight decoder is composed of two deconvolution blocks. Each deconvolution block contains a deconvolution layer of feature size 192 and kernel size 4. It is followed by a 2D batch normalization to make the training faster and a ReLU activation function. The encoder network ends with a convolution layer of feature size K+1 and kernel size 1. It generates heat maps from the encoder features at 1/4 of the input image resolution to limit the number of parameters of the model. The total number of parameters of our model is 7.6 millions.

#### 3.1.3. Training

The heat maps generated by our model provide a softmax score for each pixel. This score classifies the pixel to either the key point class or the background class. We use a cross-entropy loss for the training of the model: (1)$$L=-\sum_{x}\sum_{k}\alpha_{k}\times t_{x,k}\times log\left(p_{x,k}\right)$$
where *x* is a pixel position at the heat map resolution, *k* is a class (one of the key point or the background), $\alpha_{k}$ is a constant weight for the class *k*, $t_{x,k}$ is the target probability of the heat map pixel at position *x* for the class *k*, and $p_{x,k}$ is its softmax score.

On ground truth heat maps, each key point corresponds to a disk with a radius of 10 pixels. The aim is to improve the network convergence by regressing large enough areas. $\alpha_{k}$, for the key point classes, is also set much higher than $\alpha_{k}$ for the background class to balance the class influence in the loss.

#### 3.1.4. Key Points Location

To locate the key points from the estimated heat maps during inference, we binarize the output heat maps by setting all values above a given threshold *t* to 1. The ke ypoint position is then computed as the center of mass of the binarized heat map. If this position is inside a border margin of width *m*, the key point is discarded as its position is most probably not accurate due to missing local context and camera lens distortion.

#### 3.1.5. Perspective-Aware Key Point Sampling

In the original approach of Nie et al. [10], the key points are regularly sampled on the pitch template. The camera’s 3D to 2D projection increases the concentration in the frame of the key points located at the opposite side of the pitch in comparison with the key points located on the camera side. Yet, the RANSAC algorithm often tends to converge to a solution estimated with selected key points close to each other in the frame space. The re-projection threshold in pixels indeed classifies inlier or outlier key points. It represents a looser constraint if the key points are more concentrated. Consequently, the RANSAC will most likely estimate the homography with key points from the camera on the opposite side of the pitch, where the resolution is lower. The accuracy of the estimated homography may not, therefore, be optimal.

Consequently, we used our previous work to sample the key points more uniformly in the frame space [92], in the pitch width between the camera side, and the opposite side. Nie et al. sample the key points uniformly in the pitch space. Ideally, we would like to sample them uniformly in the frame space. This is, however, impossible, since the purpose of our system is to estimate the homography between the pitch and frame spaces. We therefore designed a perspective-aware grid sampling in the pitch space that, on average, produces more uniform results in the frame space. We model $w_{i}$ the *i*th distance between the key points in the court width axis and starting from the camera side using an arithmetic progression
(2)$$w_{i}=w_{0}+i\times r$$
with $w_{0}$ being the distance in the court width axis between the first two keypoints nearest to the camera and *r* being the common difference.

Considering the sum of an arithmetic progression, we therefore have the following: (3)$$W=\left(N-1\right)\frac{2\times w_{0}+\left(N-2\right)\times r}{2}\;\mathrm{for}\;N\geq 2$$
with *W* being the real-world width of the court and *N* being the total number of key points on the court width axis. We calculate *r* by setting a value to $w_{0}$ with the following formula: (4)$$r=\frac{2}{N-2}\left(\frac{W}{N-1}-w_{0}\right)$$

The differences between the uniform grid sampling and the perspective-aware grid sampling methods are illustrated in Figure 4.

#### 3.1.6. Self-Supervised Pre-Training

Initializing an encoder with pre-trained weights allows for faster convergence as well as for an improvement in performance in the case where the pre-training domain is larger and close enough to the domain of the task at hand. In recent years, Self-supervised Learning has received a lots of interest due to its ability to learn a good representation from a large source of unlabeled data, more specifically, on transformers [93,94]. A pretext task is designed to perform label-free supervision.

We pre-trained our encoder on the training split of the SoccerNet-v2 dataset [1] for action spotting, which contains 300 football matches that last around 90 min each. The videos are the live broadcast version of the matches. The on-going matches, with the points of view of the players on the field, the trainers, spectators, and the players on the bench, are shown. Also, during a match, different actions occur, such as shots and passes, at different positions on the field. This semantic diversity explains why contrastive learning [95,96,97] is suitable for learning representation in this dataset. It creates pairs of positive views from images using data augmentation and brings closer the latent representation of the positives while pushing the representation of other images.

We used our recently introduced soft contrastive learning pretext task called SCE [98]. In SCE, the objective function also maintains relations computed from one view, which have similarities between representation in the latent space for the other view. It has the advantage of speed in the convergence of contrastive learning to perform shorter training.

The dataset is extracted at 2 Fps which creates about 3.3 M images. This large number of images from a domain close to our task of interest provides better-suited initialized weights than an ImageNet supervised pre-trained encoder, as shown in Section 4.1.5.

#### 3.1.7. Data Augmentation during Training

Data augmentation allows us to virtually increase the number of camera setups and therefore improve the model generalization. Thus, we use horizontal flips to randomly exchange the sides of both teams. We also crop the training images with a random scale factor varying from *s* to 1 and resize them to the original input size. Some random erasing is used to simulate the occlusions of the soccer pitch by the players and encourage the network to use all the visible image features to produce robust results. We replace the pixel three color channel values of several small random areas of the image by 127.5.

### 3.2. Player Tracking

#### 3.2.1. Overview

Previous player tracking methods have a tendency to generate many identity switches when players occlude each other. In order to generate accurate individual trajectories, we designed our method to avoid switches. Unlike previous approaches, where spatial and re-ID criteria are generally used together to match detections to existing tracks, our method employs them in two distinct steps. Our re-ID model is fine-tuned in a self-supervised manner using the player thumbnails from the processed clip, as opposed to the classical approach of using a frozen model. This approach enables us to better recognize the fine identification details of the players.

Our 2D player tracking framework follows the tracking by detection paradigm. It is an offline method as it requires the full video sample to optimize player tracks. The first step is to detect the players visible on the field. We use, for this purpose, a state-of-the-art detector: YOLOX [84]. The second step generates player tracklets by chronologically processing the video frames. Intersection-Over-Union (IoU) and distance criteria are used to measure the similarity between new detections and existing tracks’ bounding boxes. They are matched with a classic bipartite matching approach [42]. The camera motion compensation reduces the number of mismatches in the case of player occlusions. In short time intervals, this quick method generates reliable tracklets, as shown in Figure 5. Ambiguities happen when some players occlude other ones. In this case, the active tracklets are stopped and new ones are started.

The third step of our method fine-tunes a pretrained re-identification (re-ID) network with a triplet loss formulation so as to later match tracklets from the same player. The network training data are the non-ambiguous player thumbnails from the previously generated tracklets. The fourth step merges the generated tracklets based on the similarity of their re-ID vectors. It also makes sure that a player does not appear twice in a video frame and that he does not teleport. The whole process is summarized in Figure 6.

#### 3.2.2. Player Detection

We train a YOLOX model [84] to detect the players on the field. It has, indeed, demonstrated in previous work its ability to well detect persons for team sport tracking applications [17,18,85]. It also achieves a good trade-off between speed and accuracy. We use a pre-trained model on the COCO dataset and fine-tune it with the tracking training set of the dataset used for testing. We use the same training parameters as Zhang et al. [44].

#### 3.2.3. Tracklet Generation

Our tracklet generation algorithm is online. It is based on the classic SORT approach [42]. The algorithm processes the frames chronologically and matches the detections of the current frame with the tracks already created at the previous frames. Bounding boxes between the current and the previous frames are associated using bipartite matching with an Hungarian algorithm [99]. The first matching is performed with IoU criteria. For the detections and tracks that remain unmatched after this first step, a second matching is performed with bounding box center distance as criteria if this distance is inferior to $c_{max}$. The detection bounding boxes that still have not been matched after this step are considered as new tracks composed of a single frame.

Being able to accurately predict the updated position of a track in the current frame is a key factor to correctly match it with the detections of the current frame. Many previous works resort to a Kalman filter [43] to predict the absolute position and size of the bounding boxes. However, in videos of team sport games, the camera moves to follow the action. In the case of dynamic camera motions, the Kalman filter has difficulties in generating accurate predictions. When there are many players in the same area, this can cause bounding box mismatches and, consequently, identity switches. To mitigate theses issues, our method estimates the camera motion between each frame as a homography. We then use a Kalman filter to predict the bounding box relative motions and size variation compensated by the camera motion. These predictions are then added to the tracklet’s last known position and size to be associated with the current frame detections by the bipartite matching algorithm. In the case of abrupt motion, the Kalman filter often fails to generate accurate predictions. It is therefore disabled if the norm of the translation component of the homography is superior to the threshold $h_{max}$.

The camera motion between two frames is estimated by first extracting good features to track [100] in the previous frame. The updated positions of these features are predicted in the current frame with the OpenCV version of the Lucas–Kanade optical flow in pyramids [101]. The final homography is computed using a RANSAC algorithm [91] with DLT [32].

We want to avoid as much as possible having several identities inside a single tracklet because the generated tracklets are later used to fine-tune the re-identification network. We detect ambiguous bounding boxes by checking if, at a given time, the IoU between two different tracklet bounding boxes exceeds a threshold τ. We split tracklets at the beginning of each continuous sequence of ambiguous bounding boxes. In this case, the current tracklets are stopped and new ones are created starting at this point in time, as illustrated in Figure 7.

#### 3.2.4. Tracklet Merging with Re-Identification

To generate full player tracks, we need to merge the tracklets from the same players. We carry out this task with appearance criteria. We assume that the players keep the same appearance during the full video sample. Since all the players from the same team wear the same jersey, identifying them from the tactical camera point of view is a hard task. We start with a re-ID model pretrained on the target domain and fine-tune it so as to learn the distinctive appearance details (hair, skin color, accessories, jersey numbers, etc.) of the players present in the video clip. This process does not require any annotation. We only use our previously generated tracks as training data.

##### Re-Identification Model Training

Our re-identification model is a Multiple Granularity Network [102] with a ResNet-50 backend [65]. The network is initialized with a checkpoint trained on the Market1501 dataset [103] and is then fine-tuned on the training set of the tracking dataset with a triplet loss formulation [104]. To build each training batch, we select positive and negative samples only from the same video sample. We follow the batch hard strategy that brings the hardest positive samples closer and pushes away the hardest negative samples.

In order to specialize the re-identification model to the target test sequence, the model is fine-tuned a second time for each video sample by using the previously generated tracklets. We use the player thumbnails of our previously generated tracklets as training data. However, to train with clean input data, we discard thumbnails that contain several players. So, when, in a given frame, player bounding boxes are marked as ambiguous (see Section 3.2.3), their respective thumbnails are not used to fine-tune the model. The tracklets containing only ambiguous thumbnails are discarded. We build each triplet batch by selecting a random player image from a given tracklet as the anchor. The $n^{+}$ positive samples are randomly selected among the same track. As it is impossible that a player appears twice in a frame, the $n^{-}$ negative samples are selected from concurrent tracklets that appear at the same time as the anchor track. If there are not enough distinct tracklet thumbnails to build a full batch with $n^{+}$ and $n^{-}$ samples, some thumbnails are included several times. Examples of the selection of an anchor, positive, and negative samples are depicted in Figure 8.

##### Iterative Merging

Once our re-identification model has been fine-tuned, we compute a re-ID vector for each tracklet by taking the mean re-ID vector of $n_{p}$ images uniformly sampled along this tracklet. By computing the Euclidean distance between all pairs of the tracklets’ re-ID vectors, we obtain a tracklet similarity matrix. The tracklets are merged using an iterative algorithm. At each step, the tracklets with the minimum re-ID distance in this matrix are selected as merge candidates. Two tracklets cannot be merged if they are visible in the same common frames. The iterations stop when the re-ID distance between the two merge candidate tracklets is above a threshold $r_{max}$. We indeed consider that, above this value, the re-ID vectors are too far away to correspond to the same player. This algorithm does not need to know the number of visible players. It merges tracklets until the stopping criteria are reached.

We also noticed that, with the way the video samples are filmed, if a player leaves the camera field of view on one side of the frame (left or right), he will most likely re-enter the field of view from the same side. In order to prevent player teleportation, we consider that two tracklets cannot belong to the same player if the distance in the frame space between the last position of the first tracklet and the first position of the second tracklet is above $d_{max}$. This criteria only apply if the time between these tracks is inferior to $t_{max}$.

### 3.3. Player Locations on the Sport Fields

Given our sport field registration method and our player tracker, we can generate the trajectories of each player on the soccer field. Our tracker in the frame space generates bounding boxes and their associated player identities. For each frame, we project the centers of the bounding boxes’ bottom edges to the pitch space thanks to the homography generated using our sport field registration method. We therefore obtain the positions of each player in the pitch space for all video frames.

## 4. Experiments

We first evaluate our sports field registration method and player tracker separately before combining the outputs of these components to determine player positions in the pitch space.

### 4.1. Sports Field Registration

#### 4.1.1. Implementation Details

Our implementation is based on the Pytorch framework. After normalization with the ImageNet mean and standard deviation values, batches of two images at a 1280×720 resolution are fed into the network during 1000 epochs for the Worldcup dataset [3] and 100 epochs for the TS-Worldcup dataset [11]. The learning rate is initially set to $10^{-4}$ and divided by 10 after two thirds of the epochs. The AdamW optimizer was used. The number of pitch keypoints *K* is set to 91. The RANSAC re-projection threshold is set to 8 pixels. We experimentally set $\alpha_{k}$ to 100 for the keypoint classes and to 1 for the background class. $w_{0}$ is set to 6.4 m (7 yards) and *m* to 4 pixels.

For data augmentations, the random image flip probability is set to 0.5. The minimum random crop scale factor *s* is set to 0.7. We also randomly erase 62 rectangles of a size of 45 by 100 pixels in order to mimic the occlusions of the soccer pitch by the players.

For the self-supervised pre-training of our encoder, we use for SCE parameters, as defined in the original paper [98], the temperatures τ=0.1, $\tau_{m}=0.07$ and the coefficient λ=0.5. The projector and predictor are a two- and three-layers Multi-Layer Perceptron (MLP) with hidden size 1024 and output size 256. For data augmentations, we also use the same ones as in the paper *strong-α* and *strong-β* and symmetrize the loss. The random aspect ratio for random resized crop is sampled between [1.33,2.21] to deal with source images of ratio 1.77:1. We use the AdamW optimizer with a batch size of 1024 and the learning rate follows a warmup over 10 epochs to reach the initial value $2\times 10^{-3}$ and decrease following a cosine scheduler to $2\times 10^{-5}$ throughout 100 epochs of training. The weight decay is set to 0.05.

The decoder layers are initialized with default Pytorch uniform distributions.

#### 4.1.2. Metrics

We use four metrics to evaluate the performance of our approach:$IoU_{whole}$ is the intersection over union of the binary mask of the whole pitch transformed by the ground truth and the estimated homographies;$IoU_{part}$ is the intersection over union of the binary mask of the visible part of the pitch transformed by the ground truth and the estimated homographies;The projection error is the average distance in meters between points randomly sampled in the frame on the visible part of the pitch and projected with the ground truth and the estimated homography;The re-projection error is the average of the distances in pixels, normalized by the frame height, between points randomly sampled on the visible part of the pitch and re-projected with the ground truth and the estimated homography.

We use the publicly available implementation of Chu et al. [11] to compute the metrics. The size of the soccer pitch used to compute the metrics is 100 by 60 m.

#### 4.1.3. Results on the WorldCup Dataset

The WorldCup dataset [3] has been used by many previous works to compare their performance. It provides soccer pitch images with their homography corresponding to the transformation between the pitch world coordinates and the 2D positions in the frame. However, its low number of training images (209) is not optimal to train deep learning models such as ours. We experimentally set the heat map binarization threshold *t* to 0.99.

We report the results obtained by our method in Table 3. It achieves state-of-the-art results for the $IoU_{part}$, $IoU_{whole}$ and projection error metrics. The projection error is the most important metric for our application since we aim to localize the pitch space in which the players were detected in the 2D frame with the highest possible accuracy. Our approach reduces it by 3.7% compared to the method of Chu et al. [11].

#### 4.1.4. Results on the TS-WorldCup Dataset

We report the results of our method and previous works on the TS-WorldCup dataset [11] in Table 4. We experimentally set the heatmap binarization threshold *t* to 0.9975. Compared to three previous works, our framework achieves the best results for all the metrics. Our method reduces the projection error by 27% compared to the method of Chu et al. [11]. All the metrics are significantly better on the TS-WorldCup dataset than on the WorldCup dataset. For example, the projection error is divided by 2.8. Our approach, therefore, benefits from the larger size of this dataset to more precisely estimate the homographies. Figure 9 shows the distribution of the projection errors and Figure 10 illustrates some qualitative results.

#### 4.1.5. Ablation Studies on the TS-WorldCup Dataset

We also perform ablation studies on the TS-WorldCup dataset to evaluate the performance of our contributions. In a first experiment, we replace the perspective-aware key point sampling using the uniform sampling used by previous works [10,11]. In another experiment, we trained our model without data augmentation. We also compare our self-supervised pre-training of the encoder on the SoccerNet action-spotting dataset with a classic supervised training on ImageNet [105] provided by the Timm project [106]. To demonstrate the ability of our vision transformer encoder to build better registration features, we replace it in two other experiments by a Resnet-18 and a Resnet-50 encoder [65], as used in previous works [10,11]. However, since these Resnet encoders output features at a spacial dimension divided by two compared to our ViT-tiny encoder, we add an additional deconvolution block to the decoder. This outputs features at the same feature dimension as the output of the encoder. We experimentally set the heat map binarization threshold *t* to 0.9975, except for the experience with the Resnet-18 encoder where it is set to 0.99. For each of these ablation experiments, the performance decreases as shown in Table 5. Among all our propositions, the data augmentation techniques have the highest impact on the performance.

#### 4.1.6. Encoder Spatial Attention Analysis

To visualize the spatial attention from the encoder ViT-tiny, we use the method proposed by reference [107]. It is interesting to look at the visualization computed on each spatial attention layer in order to analyse individual activation. The twelve spatial attention features are represented in Figure 11. We can see that the encoder rapidly focuses on the lines and edge of the field to finally concentrate its attention on the final key points’ localisation. Figure 12 represents how the attention flows from the start to the end throughout the encoder. The spatial attention values are higher near and along the lines of the field, which means that the encoder clearly relies on these lines to estimate the final key points heat maps.

### 4.2. Player Tracking

#### 4.2.1. Implementation Details

Our tracking method is implemented using the Pytorch framework. Our re-identification network is provided by the FastReID project [108]. It takes as input player thumbnails at a 128 × 384 resolution. It is trained over 100 epochs on the tracking training dataset and fine-tuned on each sample over 10 epochs with the AdamW optimizer. The learning rate is set to $1\times 10^{-4}$ and is divided by 10 for the last third of the epochs. The weight decay is set to $10^{-4}$. To build the triplet loss batches, we set $n^{+}$ and $n^{-}$ to 8. We experimentally set the parameters τ, $n_{p}$, $r_{max}$, $c_{max}$, $d_{max}$, $h_{max}$, and $t_{max}$ to 0.7, 20, 42, 30 pixels, 10 pixels, 1500 pixels, and 2 s.

#### 4.2.2. Metrics

We evaluate the performance of our 2D tracker using five commonly used metrics. The Higher-Order Tracking Accuracy (HOTA) [109] aims at quantifying the detection, association, and location performance. DetA is the detection accuracy while AssA is the association accuracy. The Multiple-Object Tracking Accuracy (MOTA) [110] focuses on the detection performance while IDF1 [111] evaluates the identity association.

#### 4.2.3. SoccerNet Tracking Dataset

The SoccerNet tracking dataset is composed of 57 soccer clips of 30 s. In the training set and 49 clips in the testing set. The clips are filmed with a single tactical view that follows the action. The annotations provide tracking data for the players, referees, and ball with no distinction of category between the persons and the ball. In our experiments, we therefore handled the ball for the detection, the re-ID, and the tracking tasks in the same way as we did the players and referees.

The first part of our experiments uses the oracle detections. In this case, we disable the ambiguous tracklet splits as our tracklet generation algorithm generates very few ID switches with the bounding boxes ground truth as input data. We performed ablation studies to measure the impact of each component on the tracking performance. The results are provided in the first part of Table 6. Figures show that camera motion compensation has the biggest impact. It increases the HOTA by 1.14 points. The teleportation checking improves the HOTA by 0.85 points. Finally, the fine-tuning of the re-identification network increases the HOTA by 0.45 points.

We also compared our player tracking algorithm with previous generic trackers [44,45,46] and an approach specialized in player tracking [85]. We also used the oracle detections in these experiments. The results are provided in the second part of Table 7. Among previous works, the C-BIoU tracker [85] obtains the best results on the SoccerNet tracking dataset. It is however, outperformed, by our approach that obtains a HOTA that is superior by 7.4 points.

Finally, we evaluated the global tracking performance of our method and previous works by using custom detections. The results are shown in the last part of Table 8. The first version of our approach was awarded first place at the SoccerNet tracking challenge in 2022 [35]. Its extension, presented in this paper, won the 2023 edition. For the experiments with BoT-SORT [47] and ByteTrack [44], since both of these methods use a YOLOX detector, we used the same weights as with our method. For the experiments with BoT-SORT, we trained the re-identification model, with the parameters proposed by the authors, with the SoccerNet tracking train dataset.

#### 4.2.4. SportsMOT Dataset

The SportsMOT dataset [19] contains 45 clips of soccer, volleyball, and basketball in the train dataset and 45 clips in the validation dataset. Unfortunately, it is not possible to evaluate our method on the test dataset since the ground truth of its 150 clips is kept private for the challenge. We report the results on the SportsMOT dataset with the custom detector of each method. For the experiments with ByteTrack [44], BoT-SORT [47], and OC-SORT [45] we used our trained YOLOX detector, since both of these methods also use YOLOX as a detector. For the experiments with BoT-SORT, we trained the re-identification model with the parameters proposed by the authors on the SportsMOT train dataset. The results in Table 9 show that our method outperforms previous works in all the metrics. It achieves a HOTA that is superior by five points compared to SportsTrack, which won the 2022 edition of the SportsMOT challenge.

### 4.3. Player Locations on the Sport Fields

To generate player locations on the pitch, we trained our sport field registration method on the WorldCup training dataset, and an additional 201 images randomly selected from the SoccerNet tracking dataset, in order to be close to the target domain. For the images from the SoccerNet tracking dataset, we manually annotated the homographies.

The qualitative results of some clips of the SoccerNet tracking dataset are presented in Figure 13 and Figure 14. Unfortunately, it was not possible to quantitatively evaluate the individual location of players since, as far as we know, there is no publicly available ground truth of such data.

This experiment utilized a single NVIDIA A100 40 Gb GPU, which was installed in a server equipped with an AMD EPYC 7543 CPU. With our current unoptimized implementation, the total execution time of our framework, for the 30 s length SNMOT-194 sequence from the SoccerNet tracking dataset, is 16 min and 21 s. A significant part of the processing time is spent in the online fine-tuning of the re-ID network (7 min and 25 s).

## 5. Discussion

We proposed a new sports field registration method and experimentally demonstrated its performance. Thanks to the usage of a vision transformer architecture, the self-supervised pre-training of our encoder, the perspective-aware key point sampling, and data augmentation techniques, our method outperforms previous works on two public datasets. Furthermore, these contributions have a low impact on the computational complexity in terms of inference time.

We also proposed an innovative player tracker in the frame space that first generates non-ambiguous tracklets with spatial criteria and then iteratively merges them according to re-ID criteria. The introduced camera motion compensation helps generate tracklets with fewer identity switches, while the self-supervised fine-tuning of the re-ID network on tracklets in each video clip allows for a better recognition of the fine identification details of the players. Our tracking approach also outperforms previous works on two public datasets and was awarded first place at the SoccerNet tracking challenge in 2022 and 2023. Our method is, however, offline. The fine-tuning of the re-ID network is, by design, computationally intensive. It minimizes the number of identity switches at the cost of longer execution times. Therefore, it currently does not fit scenarios where position results must be delivered in real time.

We conducted a qualitative evaluation of our complete framework using soccer video clips. In this context, we generated player trajectories on the soccer pitch and noticed the consistency of the results. A quantitative evaluation will, however, be needed to validate the full system. To accomplish this task, we would require a dataset that combines video clips with ground truth player positions on the pitch. To the best of our knowledge, no such publicly available dataset currently exists. We would like to point out that, although all these experiments were conducted on soccer data, the generic nature of our framework makes it suitable for any team sport video shot with a single moving camera.

## 6. Conclusions

In this article, we described a new framework for individually locating soccer players from broadcast-like videos. The framework comprises a new sports field registration method and a new player tracker within the frame space. Both components outperform the current state-of-the-art on public datasets. The projection of player positions, individually tracked in the frame space, using the homography obtained through our sport field registration method, enables us to derive player trajectories in the pitch space.

As part of our future work, we intend to perform a quantitative evaluation of player positions on the pitch. We also wish to extend our method to identify and localize the players throughout an entire game. An additional challenge in this scenario is managing player substitutions. We believe that recognizing jersey numbers could significantly enhance player identification. This could be approached as a classification problem, where each player thumbnail is assigned labels corresponding to numbers between 0 and 99, and an extra class for cases where the number is not visible. In addition, incorporating the temporal dimension along the tracklets could contribute to enhancing the robustness of jersey numbers recognition. Our iterative algorithm, which currently merges tracklets based on a re-ID criteria, could be extended to incorporate a new prioritized criteria based on jersey numbers. Finally, we would also like to test our approach on more challenging sports, such as rugby, where player occlusions are even more frequent.

## Figures and Tables

**Figure 1 sensors-23-07938-f001:**
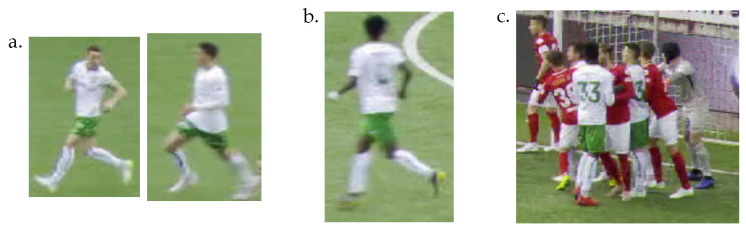
Illustration of some challenges of soccer player tracking from the SoccerNet tracking dataset [1]. (**a**) Two different players at low resolution with a very similar appearance. The details of their faces are not visible. (**b**) The low resolution of the player hinders jersey number recognition. (**c**) Some play events generate many player occlusions.

**Figure 2 sensors-23-07938-f002:**
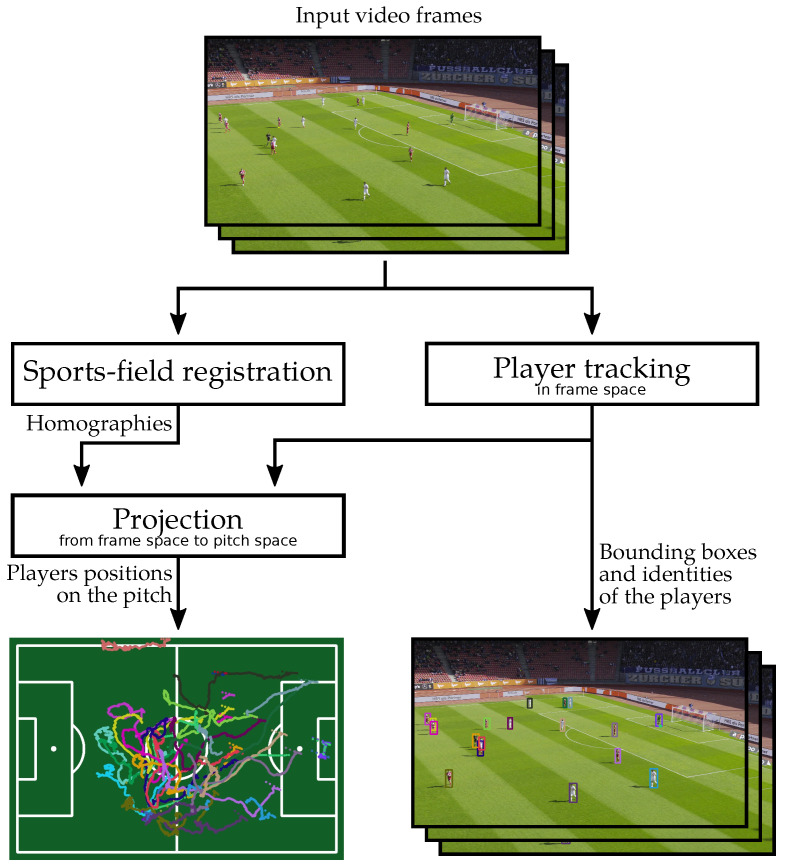
Overview of our player locations framework. It takes video frames as input and outputs individual player positions on the pitch and player bounding boxes in the frames.

**Figure 3 sensors-23-07938-f003:**
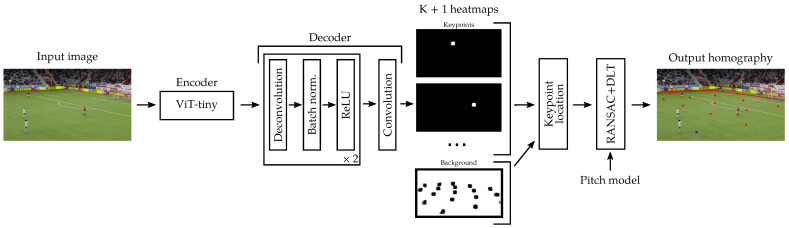
Our sports field registration approach. At inference time, the encoder–decoder model generates score heat maps for the key points and the background. The encoder is a standard ViT-tiny model [14]. The decoder is composed of two deconvolution blocks of feature size 192 and a convolution of feature size K+1. After the key points’ locations in the image space have been retrieved, a RANSAC algorithm with DLT estimates the homography between the pitch world coordinates and the 2D positions in the frame.

**Figure 4 sensors-23-07938-f004:**
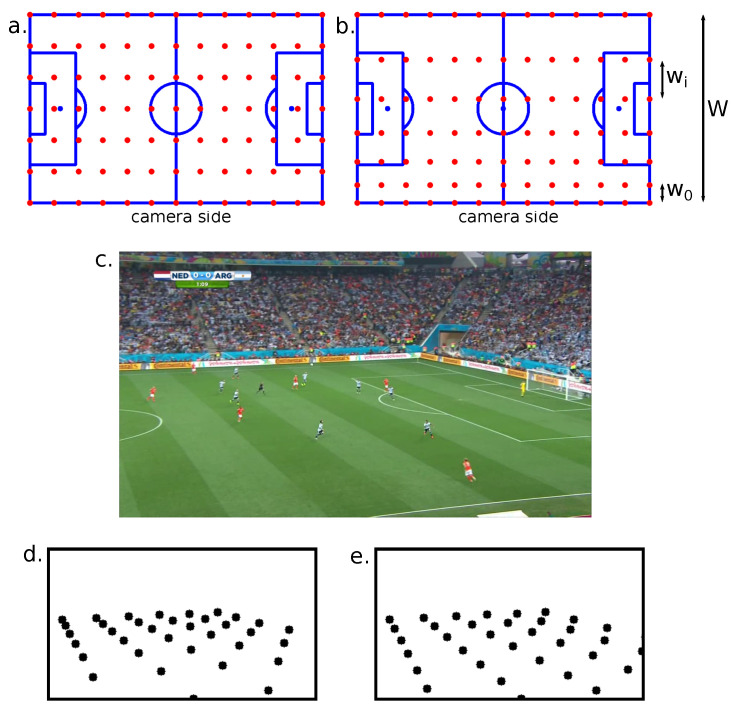
(**a**) Uniform grid sampling of the key points. (**b**) Perspective-aware grid sampling of the key points. The further the key points are from the camera, the bigger the distance between them in order to get a more uniform distribution of the key points in the frame. (**c**) Input image. (**d**) Key points ground truth obtained with a uniform sampling. (**e**) Key points ground truth obtained with the perspective-aware sampling.

**Figure 5 sensors-23-07938-f005:**
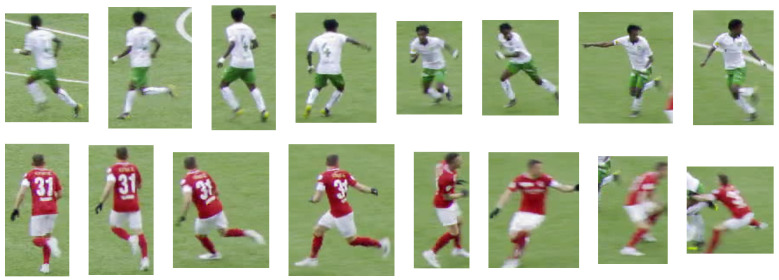
Thumbnail examples of two tracklets from the SoccerNet tracking dataset [1].

**Figure 6 sensors-23-07938-f006:**
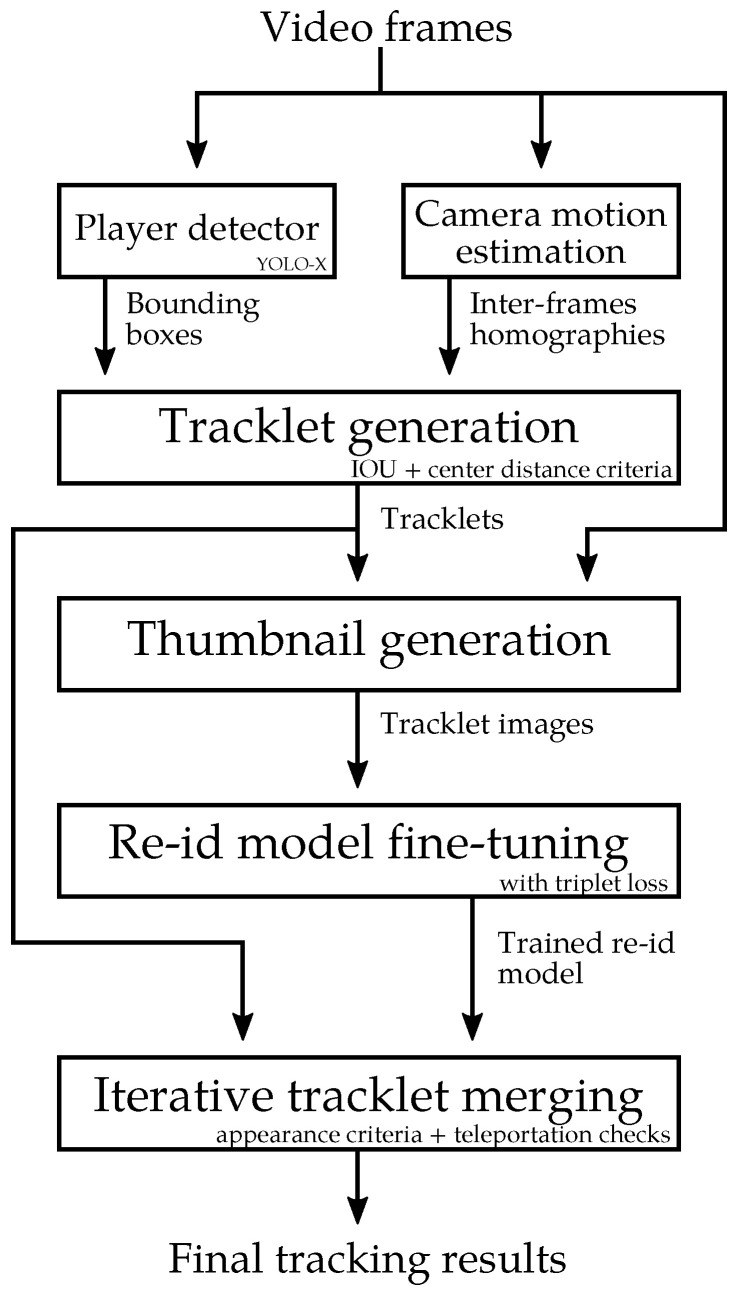
Overview of our player tracking framework in the frame space.

**Figure 7 sensors-23-07938-f007:**
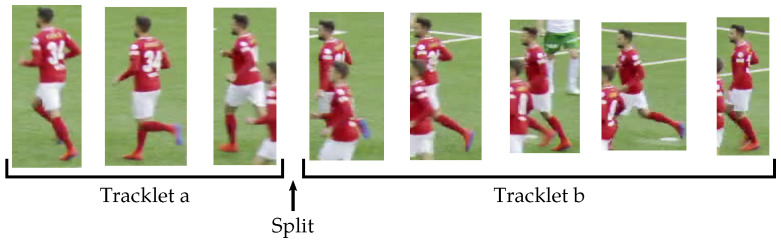
Illustration of a tracklet split from the SoccerNet tracking dataset [1]. At the beginning of the ambiguous zone, tracklet *a* is stopped and tracklet *b* is started.

**Figure 8 sensors-23-07938-f008:**
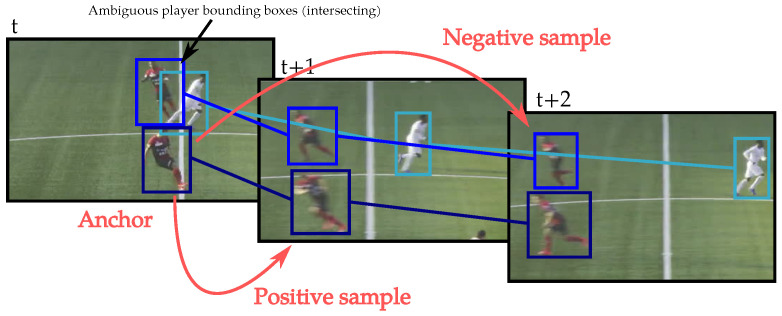
Example of the selection of an anchor, positive, and negative samples for the triplet loss used in the re-ID network fine-tuning. The links between the bounding boxes represent the concurrent generated tracks.

**Figure 9 sensors-23-07938-f009:**
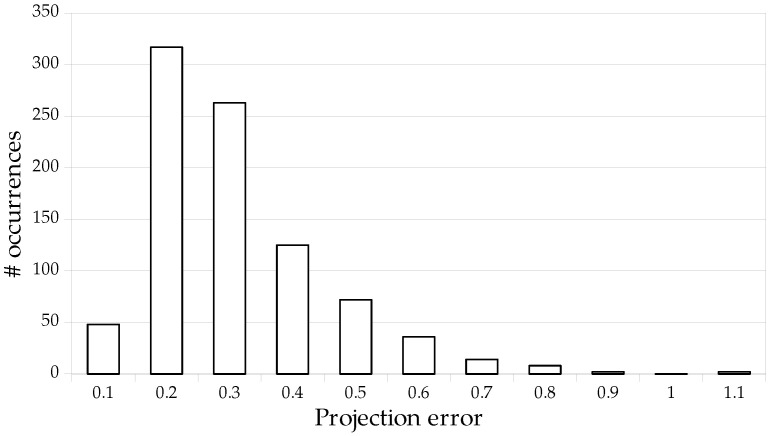
Distribution of the projection errors for each image of the TS-WorldCup dataset [11].

**Figure 10 sensors-23-07938-f010:**
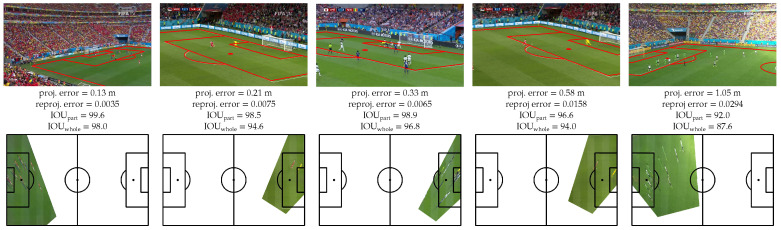
Qualitative results of our sport-field registration method on the TS-Worldcup dataset [11].

**Figure 11 sensors-23-07938-f011:**
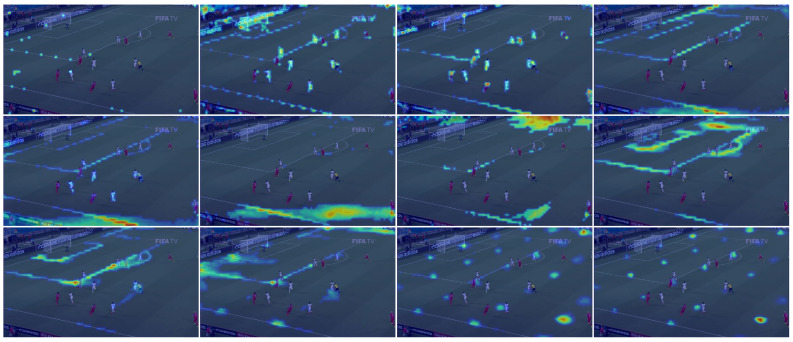
Visualization of the 12 spatial attention features from the encoder ViT-tiny. Feature from the 1st spatial attention layer at the top left, feature of the 12th layer at the bottom right. The encoder focuses on the lines and edge of the field to finally concentrate its attention on the final key points’ localisation. Image from the TS-WorldCup dataset [11].

**Figure 12 sensors-23-07938-f012:**
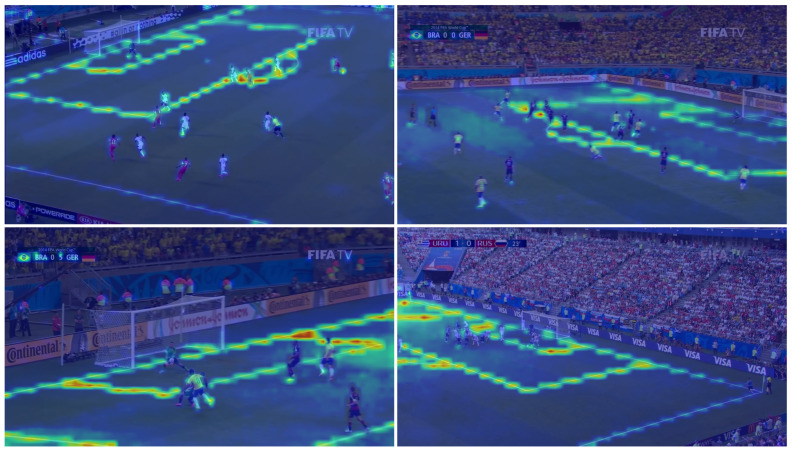
Visualisation of the ViT-tiny encoder attention rollout. Spatial attention values are higher near and along the lines of the field. Images from the TS-WorldCup dataset [11].

**Figure 13 sensors-23-07938-f013:**
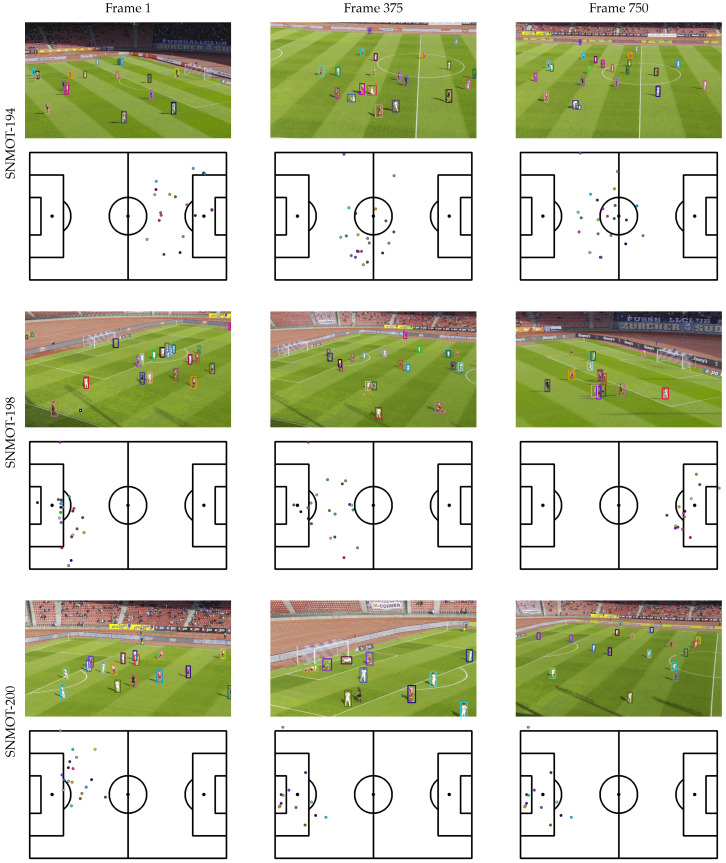
Individual location of players at specific times of some video samples of the SoccerNet tracking dataset [1].

**Figure 14 sensors-23-07938-f014:**
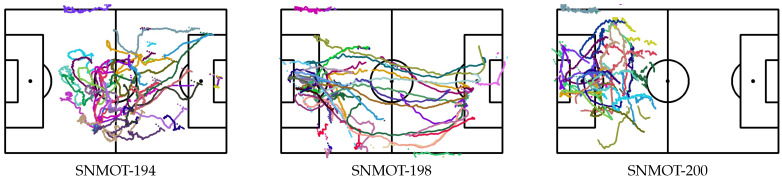
Individual location of players on some video samples of the SoccerNet tracking dataset [1].

**Table 1 sensors-23-07938-t001:** Public sport field registration datasets.

Dataset	Sport	# Frames
WorldCup [8]	soccer	395
TS-WorldCup [3]	soccer	3812
SoccerNet calibration [35]	soccer	20,028
DeepSportRadar [36]	basketball	728
RegiSwim [12]	swimming	503

**Table 2 sensors-23-07938-t002:** Public team sport player tracking datasets.

Dataset	Sport	Annotated Duration
APIDIS [89]	basketball	7 × 60 s.
FIGC-CNR [1]	soccer	6 × 104 s.
SoccerNet tracking [35]	soccer	200 × 30 s.
SportsMOT [19]	basketball, volleyball, soccer	240 × ≈19 s.
Rugby sevens [82]	rugby sevens	3 × 40 s.

**Table 3 sensors-23-07938-t003:** Experimental results on the WorldCup dataset [3]. Our method outperforms previous works in 6 out of 8 metrics.

Method	$IoU_{whole}↑\;\left(\%\right)$		$IoU_{part}↑\;\left(\%\right)$		proj. Error ↓ (m.)		re-proj. Error ↓	
	Mean	Median	Mean	Median	Mean	Median	Mean	Median
Homayounfar et al. [3]	-	-	83	-	-	-	-	
Sharma et al. [4]	-	-	91.4	92.7	-	-	-	-
Sha et al. [5]	83.2	84.6	94.2	95.4	-	-	-	-
Jiang et al. [6]	89.8	92.9	95.1	96.7	1.21 *	0.74 *	**0.017** *	**0.012** *
Citraro et al. [7]	90.5	91.8	-	-	-	-	0.018	0.012
Chen and Little. [8]	89.4	93.8	94.5	96.1	-	-	-	-
Zhang and Izquierdo [9]	91.4	**94.1**	95.9	97.3	-	-	-	-
Nie et al. [10]	91.5	93.3	95.8	97.2	0.82	0.61	0.019	0.015
Chu et al. [11]	91.2	93.1	96.0	97.0	0.81	0.63	0.019	**0.014**
Jacquelin et al. [12]	81.2	86.0	94.6	95.9	-	-	-	-
Theiner and Ewerth [13]	-	-	95.3	96.6	-	-	-	-
Ours	**92.0**	**94.1**	**96.3**	**97.4**	**0.74**	**0.55**	0.018	0.014

* Results reported by Chu et al. [11].

**Table 4 sensors-23-07938-t004:** Experimental results on the TS-WorldCup dataset [11]. Our method outperforms previous works on all metrics.

Method	$IoU_{whole}\;↑\;\left(\%\right)$		$IoU_{part}\;↑\;\left(\%\right)$		proj. Error ↓ (m.)		re-proj. Error ↓	
	**Mean**	**Median**	**Mean**	**Median**	**Mean**	**Median**	**Mean**	**Median**
Chen and Little [8] *	90.7	94.1	96.8	97.4	0.54	0.38	0.016	0.013
Nie et al. [10] *	92.5	94.2	97.4	97.8	0.43	0.38	0.011	0.010
Chu et al. [11] *	94.8	95.4	98.1	98.2	0.36	0.33	0.009	0.008
Ours	**95.7**	**96.2**	**98.3**	**98.5**	**0.26**	**0.23**	**0.008**	**0.006**

* Results reported by Chu et al. [11].

**Table 5 sensors-23-07938-t005:** Ablation studies on the TS-WorldCup dataset [11]. The SoccerNet pre-training stands for our self-supervised pre-training on the SoccerNet action-spotting dataset, while the ImageNet pre-training stands for a supervised pre-training on the ImageNet dataset. The metrics show that the ViT-tiny encoder, the self-supervised pre-training, the data augmentation and the perspective-aware key-point sampling improve the homography estimation performance.

Encoder	Encoder	Data	Keypoint	$IoU_{whole}\;↑\;\left(\%\right)$		$IoU_{part}\;↑\;\left(\%\right)$		proj. Error ↓ (m.)		re-proj. Error ↓	
**Architecture**	**Pretraining**	**aug.**	**Sampling**	**Mean**	**Median**	**Mean**	**Median**	**Mean**	**Median**	**Mean**	**Median**
ViT-tiny	SoccerNet	yes	p.-a.	**95.7**	**96.2**	**98.3**	**98.5**	**0.26**	**0.23**	**0.008**	**0.006**
ViT-tiny	SoccerNet	yes	uniform	95.0	95.6	97.9	98.3	0.32	0.26	0.009	0.007
ViT-tiny	SoccerNet	no	p.-a.	92.1	94.0	96.7	97.8	0.48	0.37	0.013	0.010
ViT-tiny	ImageNet	yes	p.-a.	95.4	95.9	98.0	98.4	0.29	0.24	**0.008**	0.007
ResNet-18	ImageNet	yes	p.-a.	94.8	95.7	97.8	98.3	0.34	0.28	0.009	0.007
ResNet-50	ImageNet	yes	p.-a.	94.6	95.7	98.0	98.4	0.32	0.26	0.009	0.007

**Table 6 sensors-23-07938-t006:** Ablation study on the SoccerNet test dataset [1] with the oracle detections. The metrics show that the camera motion compensation, the re-ID network fine-tuning, and the teleportation checks improve the tracking performance.

Re-ID Fine-Tuning	Motion Comp.	Teleport Checks	HOTA ↑	DetA ↑	AssA ↑	MOTA ↑	IDF1 ↑
yes	yes	yes	**96.57**	**99.65**	**93.60**	**99.91**	**96.16**
no	yes	yes	96.12	99.70	92.98	99.90	95.47
yes	no	yes	95.43	99.42	91.61	99.80	94.89
yes	yes	no	95.72	99.67	91.92	99.90	94.79

**Table 7 sensors-23-07938-t007:** Tracking results on the SoccerNet test dataset [1] with the oracle detections. Our method outperforms previous generic and sports-targeted people trackers on all metrics.

Method	HOTA ↑	DetA ↑	AssA ↑	MOTA ↑	IDF1 ↑
Ours	**96.57**	**99.65**	**93.60**	**99.91**	**96.16**
C-BIoU tracker [85]	89.2	99.4	80.0	99.4	86.1
OC-SORT [45] ^2^	82.0	98.6	67.9	98.3	76.3
ByteTrack [44] ^1^	71.5	84.3	60.7	94.6	-
DeepSORT [46] ^1^	69.6	82.6	58.7	94.8	-

^1^ Results reported by Cioppa et al. [1]. ^2^ Results reported by Yang et al. [85].

**Table 8 sensors-23-07938-t008:** Tracking results on the SoccerNet test dataset [1] with our custom detector. Our method outperforms previous generic people trackers on all metrics.

Method	HOTA ↑	DetA ↑	AssA ↑	MOTA ↑	IDF1 ↑
Ours	**73.29**	**73.26**	**73.42**	**87.74**	**90.12**
OC-SORT [45]	62.05	71.99	53.62	87.12	70.34
ByteTrack [44]	61.05	70.44	53.04	86.92	71.44
BoT-SORT [47]	61.12	65.44	57.16	94.63	77.92

**Table 9 sensors-23-07938-t009:** Tracking results on the SportsMOT validation dataset [19] with the custom detections. Our method outperforms previous generic and sports-targeted people trackers on all metrics.

Method	HOTA ↑	DetA ↑	AssA ↑	MOTA ↑	IDF1 ↑
Ours	**86.04**	**85.80**	**86.354**	**94.56**	**95.33**
SportsTrack [18]	80.86	81.29	80.47	-	-
BoT-SORT [47]	73.91	84.21	64.94	94.63	77.92
OC-SORT [45]	70.44	84.25	58.99	93.32	72.52
ByteTrack [44]	64.31	76.08	54.39	93.27	73.98

## Data Availability

All the video and image data used in this study are publicly available. The annotations are also publicly distributed, except the 201 homographies from the SoccerNet tracking dataset that we manually annotated.

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
