# Peer review of "Individual Locating of Soccer Players from a Single Moving View"

_sensors, 2023, doi:10.3390/s23187938_

Round 1

Reviewer 1 Report

In this paper, the authors offered a method estimates the positions of key points sampled on the pitch thanks to an encoder-decoder architecture. The attention mechanisms of the encoder based on a vision transformer captures characteristic pitch features globally in the frames.

The article can be considered for publication after addressing the following comments:

Extend the introduction to cover more findings.

Include all images in eps format for better resolution and better presentation.

Explain the definition of L in Page 6.

Comment on the results in Table 2.

Extend the conclusion to cover future extensions of the work.

proofread the whole text.

Minor editing of English language required

Author Response

Dear Reviewer,

First of all, we would like to thank you for your helpful comments which have enabled us to improve the quality of the paper. We have tried to address all of them. Please note that we changed the outline of the article as requested by reviewer 3.
Here are some answers to your remarks:

> Extend the introduction to cover more findings.

The introduction has been extended to better present our contributions and justify them.

> Include all images in eps format for better resolution and better presentation.

We have checked that all Figures were exported in a vector format. Please note that if some soccer images may appear pixelated, it is because of the original resolution of the images in the dataset.

> Explain the definition of L in Page 6.

We have fixed the definition of L, which contained an error.

> Comment on the results in Table 2.

We have added a small comment in the label of the Table 2 (now Table 3)

> Extend the conclusion to cover future extensions of the work.

We have added a new "Discussion" section, rewrote our conclusion and extended the future works part.

> proofread the whole text.

We have proofread the article and have corrected minor errors in the writing.

Reviewer 2 Report

The manuscript “Individual locating of soccer players from a single moving view” uses image processing technology and AI algorithms to locate and identify the players.

First, I would like to congratulate the authors, for the work done. The manuscript is well written, and very detailed, including a good, extensive, and recent review of similar works. The framework figures are a good contribution to the readability of the text.

The presented results outperform the public algorithms, being therefore a great contribution to improve this field. It is therefore a great reference for future works.

The authors don’t refer to the computational power involved in the processing of the data. This information could be useful as some end-users may prefer fast algorithms with less accurate results, while others may prefer the opposite.

Line 11 – IoU is only defined in line 371, while being used alongside the text

Line 38 – the text refers to the figure 2 - some of these challenges for soccer before referring to figure 1. Please rearrange the figures/numbering. In addition, figure 1 should appear after being referred in the text

Line 74 – repeated word The

Line 202 The > the

Line 214 – Equations are not numbered.

Line 219 – n is not defined. Is it equal to N??

Line 259-260, please rephrase

Line 267 – is > was

Line 281-line 282  – 2x10-3 / 2x10-5 as to be coherent with other references along the text

Line 311 – binarization > binarization

Fig 5 – change the ,  to . Also check the font to be coherent with the text

Line 361 – form or from?

Line 514 – is > are

Line 630 – I didn’t understand “clips since it is reserved for the challenge”. What do you mean?

Author Response

Dear Reviewer,

First of all, we would like to thank you for your helpful comments which have enabled us to improve the quality of the paper. We have tried to address all of them. Please note that we changed the outline of the article as requested by reviewer 3. Here are some answers to your remarks:

> The authors don’t refer to the computational power involved in the processing of the data. This information could be useful as some end-users may prefer fast algorithms with less accurate results, while others may prefer the opposite.

We have added in the section 4.3, some information about the processing time and the computational power involved. We have also added discussions about the speed of our method in the new section 5 "Discussion".

> Line 11 – IoU is only defined in line 371, while being used alongside the text

Fixed.

> Line 38 – the text refers to the figure 2 - some of these challenges for soccer before referring to figure 1. Please rearrange the figures/numbering. In addition, figure 1 should appear after being referred in the text

Fixed.

> Line 74 – repeated word The

Fixed.

> Line 202 The > the

Fixed.

> Line 214 – Equations are not numbered.

Fixed.

> Line 219 – n is not defined. Is it equal to N??

Fixed. n had to be replaced by N.

> Line 259-260, please rephrase

This discussion has been rephrased and moved to section 2.1.3.

> Line 267 – is > was

Fixed.

> Line 281-line 282  – 2x10-3 / 2x10-5 as to be coherent with other references along the text

Fixed.

> Line 311 – binarization > binarization

Fixed.

> Fig 5 – change the ,  to . Also check the font to be coherent with the text

Fixed.

> Line 361 – form or from?

We have rephrased this sentence to make it clearer.

> Line 514 – is > are

Fixed.

Line 630 – I didn’t understand “clips since it is reserved for the challenge”. What do you mean?

We mean that the ground truth of the challenge set is kept private. We have rephrased this sentence to make it clearer.

Reviewer 3 Report

I have now completed my review of " Individual locating of soccer players from a single moving view" and some recommendations were given below:

1. In the second part of the study, studies in the literature related to this subject are included. However, the novelty in this study should be revealed more clearly, what has been done that is not in the previous work or what problem the algorithm discussed in this study solves or eliminates should be clearly revealed. The originality of the work should be explained more exactly by comparing the previous work's results.

2. "Our sports-field registration method is based on the framework of Nie et al. 2021" was expressed. As stated above, it should be explained why the algorithm of Nie et al. needs to be improved and what kind of improvement is expected accordingly. This improvement should also be declared in the discussion section.

3. In Line 38, “The Figure 2 illustrates some of these challenges for soccer.” is written. It is not appropriate to go directly to Figure 2 before mentioned the Figure 1.

4. It is written as “3.8 Experiments” section. However, it is more appropriate to treat the most important part of the study, the Experiment part, as an independent section.

5. Similarly, Section 3.8.4.” was organized as “Quantitative results”. In my opinion, this should be considered as a main section where the results are examined comprehensively.

6. After the Experiments, there is a section called "4 Previous work on player tracking". This section should either be incorporated into the Experiment chapter in a different way, or it should be placed earlier. It is not clear why the previous work was previously discussed in section 2 and then again here.

7. After "3.8 Experiments" there is a new "5.6 Experiments" under the heading "5. Player tracking". In this article structure, it is very difficult to follow the work and understand what is being done. Could you please simplify your paper and make it easier to read and understand (e.g. 1. Introduction; 2. Description of Your Methodology; 3. Experiments; 4. Discussion and 5. Results or so on).

8. There should be a Discussion section where all the applications made within the scope of the study are discussed and the findings are examined in detail.

9. Conclusion must be reviewed and re-written.

10. Briefly, in this paper, a framework was described in order to extract the individual players positions from a broadcast-like video of a soccer game. Obviously, this is a good study with a strong scientific basis. However, it would be appropriate to revise the study by considering the issues mentioned above. Especially the research should be re-designed appropriately. The experimental methods should be described comprehensively.

I sincerely congratulate the academics who carried out this labor-intensive study. I hope that my constructive suggestions will contribute to the improvement of the study.

Author Response

Dear Reviewer,

First of all, We would like to thank you for your helpful comments which have enabled us to greatly improve the quality of the paper. We have tried to address all of them. Please note that we changed the outline of the article as you requested. Here are some answers to your remarks:

> 1. In the second part of the study, studies in the literature related to this subject are included. However, the novelty in this study should be revealed more clearly, what has been done that is not in the previous work or what problem the algorithm discussed in this study solves or eliminates should be clearly revealed. The originality of the work should be explained more exactly by comparing the previous work's results.

We have extended our introduction to better justify why we developed a new player tracking approach and what it brings to the table compared to the state of the art. We have also added some elements at the beginning of section 3.2.1. Our findings are also listed in the new section 5 "Discussion".

> 2. "Our sports-field registration method is based on the framework of Nie et al. 2021" was expressed. As stated above, it should be explained why the algorithm of Nie et al. needs to be improved and what kind of improvement is expected accordingly. This improvement should also be declared in the discussion section.

We have extended the introduction to better underline and justify our contributions in our new sports-field registration method. We have also added some elements at the beginning of section 3.1.1. Our contributions are summarized in the new section 5 "Discussion".

> 3. In Line 38, “The Figure 2 illustrates some of these challenges for soccer.” is written. It is not appropriate to go directly to Figure 2 before mentioned the Figure 1.

This has been fixed.

> 4. It is written as “3.8 Experiments” section. However, it is more appropriate to treat the most important part of the study, the Experiment part, as an independent section.
> 5. Similarly, Section 3.8.4.” was organized as “Quantitative results”. In my opinion, this should be considered as a main section where the results are examined comprehensively.
> 6. After the Experiments, there is a section called "4 Previous work on player tracking". This section should either be incorporated into the Experiment chapter in a different way, or it should be placed earlier. It is not clear why the previous work was previously discussed in section 2 and then again here.
> 7. After "3.8 Experiments" there is a new "5.6 Experiments" under the heading "5. Player tracking". In this article structure, it is very difficult to follow the work and understand what is being done. Could you please simplify your paper and make it easier to read and understand (e.g. 1. Introduction; 2. Description of Your Methodology; 3. Experiments; 4. Discussion and 5. Results or so on).

We have changed the outline of the article to follow the following standard :
1. Introduction; 2. Previous works; 3. Method; 4. Experiments; 5. Discussion; 6. Conclusion.
The section 2, 3 and 4 contains successive subsections about sports-field registration and player tracking. We hope this makes the structure of the article easier to read and to understand.

8. There should be a Discussion section where all the applications made within the scope of the study are discussed and the findings are examined in detail.

We have added a new section 5 "Discussion" where we list our findings and also discuss some limitations.

9. Conclusion must be reviewed and re-written.

We have rewritten the conclusion and extended the part about future works.

Round 2

Reviewer 3 Report

I believe that all the issues I have mentioned have been taken into consideration and the necessary arrangements have been made. I think that the study is more understandable and easy to follow in this form. I congratulate the authors for this beautiful work, which I anticipate will contribute to the literature.